# Analysis of Respiratory Sinus Arrhythmia and Directed Information Flow between Brain and Body Indicate Different Management Strategies of fMRI-Related Anxiety

**DOI:** 10.3390/biomedicines11041028

**Published:** 2023-03-27

**Authors:** Beate Rassler, Katarzyna Blinowska, Maciej Kaminski, Gert Pfurtscheller

**Affiliations:** 1Carl-Ludwig-Institute of Physiology, University of Leipzig, 04103 Leipzig, Germany; 2Nalecz Institute of Biocybernetics and Biomedical Engineering, Polish Academy of Sciences, 02-109 Warsaw, Poland; katarzyna.blinowska@fuw.edu.pl; 3Faculty of Physics, University of Warsaw, 02-093 Warsaw, Poland; maciek.kaminski@fuw.edu.pl; 4Institute of Neural Engineering, Graz University of Technology, 8010 Graz, Austria; pfurtscheller@tugraz.at

**Keywords:** respiratory sinus arrhythmia, directed information flow, neural pacemaker-like activity, fMRI-related anxiety, anxiety management, causal coupling, breathing rhythm, neural BOLD oscillations, cardio-respiratory coupling, breathing-entrained oscillations

## Abstract

Background: Respiratory sinus arrhythmia (RSA) denotes decrease of cardiac beat-to-beat intervals (RRI) during inspiration and RRI increase during expiration, but an inverse pattern (termed negative RSA) was also found in healthy humans with elevated anxiety. It was detected using wave-by-wave analysis of cardiorespiratory rhythms and was considered to reflect a strategy of anxiety management involving the activation of a neural pacemaker. Results were consistent with slow breathing, but contained uncertainty at normal breathing rates (0.2–0.4 Hz). Objectives and methods: We combined wave-by-wave analysis and directed information flow analysis to obtain information on anxiety management at higher breathing rates. We analyzed cardiorespiratory rhythms and blood oxygen level-dependent (BOLD) signals from the brainstem and cortex in 10 healthy fMRI participants with elevated anxiety. Results: Three subjects with slow respiratory, RRI, and neural BOLD oscillations showed 57 ± 26% negative RSA and significant anxiety reduction by 54 ± 9%. Six participants with breathing rate of ~0.3 Hz showed 41 ± 16% negative RSA and weaker anxiety reduction. They presented significant information flow from RRI to respiration and from the middle frontal cortex to the brainstem, which may result from respiration-entrained brain oscillations, indicating another anxiety management strategy. Conclusions: The two analytical approaches applied here indicate at least two different anxiety management strategies in healthy subjects.

## 1. Introduction

General Background: The term “respiratory sinus arrhythmia” (RSA) denotes a physiological phenomenon based on entrainment between cardiovascular and breathing rhythms. Typical RSA is characterized by a shortening of cardiac beat-to-beat intervals (RRI), i.e., increase in heart rate (HR) during inspiration and RRI prolongation (HR decrease) during expiration [1,2]. RSA is the main part of high-frequency (HF) heart rate variability (HRV) and is often termed as HRV in synchrony with respiration [3,4]. For a long time, RSA has been considered to represent respiration-driven vagally mediated HR modulation [5], thus serving as an index of parasympathetic tone. The low-frequency (LF) component is considered to be a product of both sympathetic and parasympathetic activities [6]. Consequently, the relation of LF to HF HRV has been used as a measure of sympatho–vagal balance [7,8,9].

However, HRV and RSA comprise much more complex interactions between the cardiovascular, respiratory, and autonomic nervous systems [3]. Besides fluctuations in vagal activity, baroreflex mechanisms and respiratory fluctuations of intrathoracic pressure are considered to be important sources of RSA [4,10,11]. Due to the close vicinity of the central respiratory pattern generator to sympatho–excitatory neurons in the rostroventrolateral reticular nucleus, respiration exerts a marked influence upon sympathetic nerve activity [12,13,14]. Sympathetic depression during inspiration and an abrupt re-increase of activity during postinspiration is a typical pattern of respiratory–sympathetic coupling [12,13], which in turn, may induce modulation of HRV. A study on cardiorespiratory coupling during sympathetic activation due to graded head-up tilt showed a reduction in the direct effect of respiration on the heart rate with progressive sympathetic activation [15]. 

As HRV reflects the autonomic control of the heart and the autonomic balance, the magnitude of RSA is a well-accepted marker of cardiovascular diseases [7]. Chronic cardiovascular diseases such as hypertension have been described to reduce the amplitude of RRI oscillations and even to abolish RSA [16,17]. Emotional disorders such as depression or anxiety have also been shown to reduce the magnitude of RSA [18,19,20]. In contrast, a higher RSA magnitude was observed in healthy individuals with elevated anxiety [21]. Similarly, recent studies on fear conditioning showed increased HF HRV, indicating increased vagal activity in the response to the conditioning in contrast to a neutral stimulus [22]. These findings indicate that HRV is more than only a marker of sympatho–vagal balance. HRV is part of a complex interaction between cortical, limbic, and autonomic structures involved in adaptive functions such as emotional learning [23,24]. Accordingly, voluntary modulation of HRV including RSA such as HRV biofeedback training can be used as a therapeutic instrument [25,26]. For instance, forced (paced) resonance breathing at about 6 breaths/min, which can be applied as a relaxation technique, is accompanied by a high HRV amplitude [27,28]. 

The specific background of this study is the observation of phase-shifts in the respiration-related RRI variations. Recently, we found an inverse phase-relation in the cardio-respiratory coupling with HR decrease (RRI increase) during inspiration and HR increase (RRI decrease) during expiration in some participants of a functional magnetic resonance imaging (fMRI) study [29]. This type of cardio-respiratory coupling termed negative RSA (nRSA) has been observed for the first time in awake and spontaneously breathing persons and occurred predominantly in participants with high anxiety [30]. Of note, in contrast to effects of cardiovascular and other disorders as mentioned above, the magnitude of respiration-related oscillations is not reduced in nRSA. For comparison of (classical) RSA and nRSA, please see Figure 1.

Prior to our observation, nRSA has been described in anesthetized patients [31] or in persons with paced slow breathing [32]. While classical RSA means a respiration-driven modulation in HR, the rather uncommon nRSA is thought to result from a change in dominance; this means that the cardiac rhythm gains dominance over the respiratory rhythm. Using a wave-by-wave analysis, we demonstrated that nRSA was mostly associated with a transient reversal in the direction of entrainment, i.e., the RRI rhythm was the leading rhythm and entrained the respiratory rhythm for a limited period of time [30]. We hypothesized that the dominance of the RRI rhythm over the respiratory rhythm might result from the activation of a central pacemaker, which is thought to be part of an anxiety management strategy. This pacemaker predominantly acts in the 0.15 Hz range and is most probably located in the brainstem [33].

Issues addressed in the present study: In our previous studies, nRSA was predominantly observed in subjects with slow RRI oscillations and slow breathing at rates around 0.10–0.15 Hz [29,30]. A majority of our subjects, however, breathed at a rather “normal” rate (0.2–0.4 Hz, centered around 0.32 Hz; further referred to as fast breathing). In these cases, phase-coupling was also detected, but there was no clear predominance of RSA or nRSA [30]. Of note, (classical) RSA and nRSA are not the only phase-relations in cardio-respiratory coupling. Breathing and RRI oscillations may adopt any other phase-relation, which is termed “indefinite RSA” [32]. The portion of indefinite RSA was higher during breathing in the 0.2–0.4 Hz band compared to slow breathing. The prevalence of indefinite RSA impedes the analysis of the linkage between cardio-respiratory coupling and anxiety management (for a more detailed discussion, please see the Limitations section). As anxious subjects often breathe at higher rates (above 0.2 Hz) [20], the question whether signals in the HF band also allow detection of a directed information flow is of particular interest. Consequently, we extended our approach in the present study by analysis of blood-oxygen-level-dependent (BOLD) oscillations in various brain regions.

The BOLD signal is a complex signal composed of three interacting components: (i) cerebral metabolic rate of oxygen consumption (CMRO2) due to neural activation, (ii) cerebral blood flow (CBF, Mayer waves) and (iii) cerebral blood volume (CBV). Changes in BOLD signals only reflect neural activity (neural BOLD) accurately if vascular components (CBF, CBV) are not altered [34]. In contrast, the classical neurovascular coupling (NVC) reflects the close temporal and regional linkage between neural activity and cerebral blood flow (CBF) [34,35], whereby cerebral autoregulation, the intrinsic dynamic ability of cerebral vessels to maintain CBF despite fluctuations in arterial blood pressure, plays a dominant role. The evidence of such blood pressure oscillations on BOLD fluctuations is not completely clear and needs further research. In this respect, studies on the interaction of cerebrovascular and cardiovascular variability are important [36]. Neural BOLD oscillations lag behind RRI oscillations, and the time delay between the two oscillations is positive (pTD). Vascular BOLD oscillations lead before RRI oscillations, and the time delay between the two oscillations is negative (nTD) [33]. Cardiovascular and respiratory functions are involved in a multitude of interactions with cortical activities including the limbic brain areas such as the amygdala and hippocampus, regions related to regulation of emotions [23,24,37,38]. The directional interactions between the heart rate and respiration revealed that the influence is much stronger from the respiratory to the cardiac rhythm than in the opposite direction [39,40,41,42,43]. The direction of coupling between the heart rate and respiration may be reversed in some pathological states or during sleep [44,45].

The above-mentioned studies considered only two signals and showed directionality in cardio-respiratory interactions, which does not necessarily mean causal relations. The interaction between multiple signals including reciprocal connections is provided by means of Directed Transfer Function (DTF), which is an extension of the Granger causality principle [46] and yields causal coupling within a set of signals [47]. A recent study analyzing information flow between RRI, respiration and BOLD signals in various brain regions of interest (ROIs) in the 0.1–0.2 Hz band demonstrated that during low/no anxiety, an information flow from the cortex to the brainstem predominated while during elevated anxiety, a flow from the brainstem to the cortex was dominant. During elevated anxiety, the RRI signal received modulating input from information flow both from the cortex and from the brainstem, supporting our assumption of a neural pacemaker involved in the processing of elevated anxiety [48]. Consequently, the evidence of strong information flow from the brainstem to the RRI rhythm in the present study might explain the enhanced dominance of the RRI rhythm over the respiratory rhythm and the induction of the phase-shift typical for nRSA.

Aims of this study: The present study is designed to investigate whether detection of directed information flow is possible with signals in the HF band and whether the DTF analysis allows a more comprehensive assessment of cardio-respiratory coupling. We would expect that quantification of the directed information flow between various brain regions and cardiac and respiratory signals may extend the information provided by the analysis of phase-relations. The specific questions of this study are: (i) Can the analysis of directed information flow provide additional support for the hypothesized neural “pacemaker” by demonstrating different information flows between RRI, respiratory and BOLD signals in situations with RSA or nRSA? (ii) Can such an analysis provide information about the flow between slow RRI and fast respiratory oscillations? (iii) Can this type of analysis provide more detailed information on strategies of anxiety regulation?

## 2. Materials and Methods

This study was based on recordings of respiration, RRI and BOLD signals and on self-assessments of state anxiety (AS) of 10 selected participants of an fMRI study. The time course and phase-relations between RRI and respiration were analyzed wave by wave to detect the occurrence of RSA/nRSA. Synchronization between RRI and BOLD waves was assessed using phase-locking statistics. Directed Transfer Function (DTF) was applied to estimate causal coupling between respiration, RRI time courses and BOLD signals.

Study approval: All participants provided written informed consent after reading the protocol of the study, which was approved by the local Ethics Committee at the University of Graz (number: GZ. 39/75/63 ex 2013/14). Our research was performed in accordance with the ethical standards laid down in the 1964 Declaration of Helsinki.

Experimental design and participants: The present work is based on data collected in a major fMRI study that was originally designed to investigate the relations between brain oscillations in various brain regions and their relation to initiation of self-paced movements. During performance and first evaluation of the experiments, the aspect of anxiety and its relation to BOLD, RRI and respiratory oscillations received increasing attention and has become the main focus of recent studies. A detailed description of the experimental procedure in relevance to the anxiety-related studies is given [33]. In brief, each participant had to perform two identical fMRI sessions separated by a break of 50 min. AS was assessed at four different stages of the experiment.

Each session lasted about 40 min and started with a first short questionnaire to assess AS, followed by the first resting state, several movement tasks, the second resting state and the second AS questionnaire. The four resting states with their related AS values were labeled R1, R2 (first session), R3 and R4 (second session). This experimental protocol was designed to assess whether the subjective perception of anxiety remains constant during a single fMRI procedure and over an examination sequence. 

In total, 23 healthy persons aged 19–34 years (12 female, 22 right-handed) participated in this experiment. They were naïve to the purpose of the study and had no former MRI experience. All of them had normal or corrected-to-normal vision and were without any record of neurological or psychiatric disorders (as assessed by self-report). From these, 10 persons with elevated anxiety were selected for the present study. A detailed description of the selection criteria is given in [30]. In brief, based on the overall mean AS values + 1 standard deviation (SD), an arbitrary threshold was defined at AS > 21. Ten participants exceeded this threshold in at least one resting state. From them, the resting states with the highest AS values were evaluated using a wave-by-wave analysis and a Directed Transfer Function calculation (for more details, see below).

Assessment of state anxiety: For assessment of anxiety, the state-trait anxiety and depression inventory (STADI; [49]) was used, which is based on the State-Trait Anxiety Inventory [50]. The STADI is an instrument constructed to assess both state and trait aspects of anxiety and depression. It allows a reasonable separation of anxiety and depression symptoms. The items were presented on a screen within the scanner and were answered via a trackball. The possible range of scores is from 10 (very low anxiety) to 40 (very high anxiety). 

Recording of respiration and ECG: For the recording of ECG and respiration during BOLD image acquisition, we used the PERU (Physiological ECG/Respiration Unit), a component of the Magnetom Skyra MR system (Siemens AG Healthcare Sector, Erlangen, Germany) made for recordings of ECG and respiration inside the scanner. Data were sampled at a rate of 400 Hz. The respiratory data were acquired using a pneumatic cushion, which was attached to the participant by a respiration belt and connected to a pressure sensor on the PERU. The ECG beat-to-beat intervals (RRI) were detected using the fMRI plug-in for EEGLAB [51] and the FASTR algorithm for removal of gradient-induced artifacts. As ECG signals recorded during MRI with a high scanning rate often have a reduced quality [52], the RRI signals were re-processed using the Kubios HRV Premium Package (Kubios Ltd., Kuopio, Finland; version 3.0.2) [53].

fMRI BOLD recording and selection of ROIs: BOLD signals reflect changes in blood oxygenation and changes in CBF accompanying neural processes such as activation of a central pacemaker. Both types of changes can be detected through BOLD contrast measurements [54,55]. Besides cortical and limbic regions, BOLD signals from the brainstem (more exactly from pons/brainstem) were of particular interest in this study as two important sources for rhythmic BOLD signals are concentrated in this region. These are, on the one hand, the basilar artery, which is close to the medulla and the pons. On the other hand, important nuclei involved in cardiac control such as the solitary tract nucleus, vagal motor neurons in the nucleus ambiguus and dorsal vagal motor nucleus are localized in the medulla and the pons [56,57].

Functional images were acquired with a 3T scanner (Magnetom Skyra, Siemens AG Healthcare Sector, Erlangen, Germany) using a multiband GE-EPI sequence [58] with a simultaneous six-band acquisition with TE/TR = 34/871 ms, 52° flip angle, 2 × 2 × 2 mm^3^ voxel size, 66 contiguous axial slices (11 × 6), acquisition matrix of 90 × 104 and a FOV of 180 × 208 mm^2^. Finally, the AAL atlas [59] was used to extract time courses of BOLD signals for specified regions of interest (ROIs) in the left pre-central gyrus (PCG, ROI 1), left middle frontal gyrus (MFG, ROI 7), left medial superior frontal gyrus (ROI 23), left hippocampus (ROI 37), left amygdala (ROI 41), left brainstem (ROI 93, ROI 103) and right brainstem (ROIs 96, 98, 100). For details see [48,60].

Wave-by-wave analysis of the phase-relations between respiration and RRI waves: The duration of breathing and RRI waves (period duration, PD) for the selected participants was detected period by period. A detailed description of this method is given in [29]. In brief, we detected the start of inspiration and expiration from the minima and maxima, respectively, in the respiration curve. From these, we calculated the period duration (PD) of each breath. Similarly, the PD of RRI waves was determined between two consecutive minima in the RRI curve. The phase-relations were measured as the time interval from the start of inspiration/expiration to the subsequent maximum and minimum of the RRI signal and were expressed in the percent of inspiration or expiration duration. For a detailed description of the criteria for classification as RSA or nRSA, please see [30]. Classical RSA is characterized by RRI maxima occurring in late expiration or early inspiration, and/or RRI minima in late inspiration or early expiration. In contrast, nRSA means that RRI maxima occur in late inspiration or early expiration, and RRI minima in late expiration or early inspiration. This method is an appropriate instrument to characterize frequency and phase coupling. Shifts in the phase-relation between the coupled rhythms indicate modulations of coupling and can give hints on the dominance of the rhythms.

Calculation of phase-locking between RRI and BOLD waves: The frequency-specific synchronization between RRI and BOLD waves as a putative mechanism of neural integration was analyzed by means of phase-locking statistics. We used the “Cross-wavelet and Wavelet Coherence” toolbox [61] and focused on the phase component for computing the phase-locking value (PLV). PLV is a normalized measure of how much the phase difference between two signals changes in a user-chosen time window, regardless of the actual phase difference value; for more details, see [62]. The PLV was calculated in the 0.05–0.15 Hz band between the RRI time course and the BOLD time series. The calculation procedure is based on a method reported in [63] and is described in detail in [64]. As a result, two PLV-based parameters were computed: (i) the time delay (TD), which is the phase delay converted to time; and (ii) the significant length of phase coupling (sigbin%), which is the percentage of time samples within the time series that survive a 0.05 significance threshold. A positive time delay (pTD) indicates that RRI oscillations precede BOLD oscillations and is typical for neural BOLD oscillations. A negative time delay (nTD) indicates that RRI oscillations lag behind BOLD oscillations and is typical for vascular BOLD signals.

Computing of directed coupling: The interaction between respiration, RRI time courses and BOLD signal was studied by means of Directed Transfer Function (DTF) [65], which allows to estimate causal coupling between the signals as a function of frequency. DTF is an extension of the Granger causality principle [46] to the multivariate case [66]. Granger causality measures the predictability for two time series. If the variance of the prediction error for the second time series is reduced by including past measurements from the first time series in the linear regression model, then the first time series can be said to cause the second time series. The Granger causality principle is equivalent to 2-channel Multivariate Autoregressive Model (MVAR) but it may be extended to an arbitrary number of channels [67,68]. For our study, MVAR was extended to 6 channels [47,65], four BOLD signals, respiration and RRI time courses. Epochs of 40 s were used for the analysis. We applied the full frequency version of DTF (ffDTF), which describes the causal influence of one signal on another one (the destination signal) at a certain frequency normalized in respect of inflows to the destination channel. In ffDTF, the normalization factor does not depend on frequency. The method is described in detail in [48,60]. The following bands were studied: 0.05–0.15 Hz, 0.1–0.2 Hz and 0.2–0.4 Hz. 

The statistical procedure of finding significant differences in average coupling values in the chosen frequency band was based on the bootstrap approach, which is described in detail in [60]. In brief, a common pool of couplings from both cases, inflows (type 1) and outflows (type 2), for a given channel was created. From this pool were randomly selected coupling values, which were marked as type 1 or type 2. Then, the differences between drawn type 1 and type 2 values were calculated. Based on about 10,000 repetitions of this procedure, we received the empirical distribution of coupling difference values corresponding to the hypothesis of no differences between inflows and outflows from the given channel. Original values of connectivity difference lying outside an assumed confidence range of 95% of such obtained distribution were considered to represent significant differences between inflows and outflows.

## 3. Results

The results from wave-by-wave analysis extended by the results of the BOLD analyses (TD, sigbin%) are summarized in Table 1. The RRI oscillations presented with two dominant frequency bands centered at ~0.1 and 0.16 Hz. On average, more than 50% of all RRI periods belonged to one of these dominant frequency bands. The RRI and respiratory signals were not homogeneous. Respiration of these subjects showed three dominant frequency bands centered at ~0.1, 0.16, and 0.32 Hz. Note, 0.16 Hz and 0.32 Hz (double of 0.16 Hz) but not 0.1 Hz are parts of the binary hierarchical model of Klimesch [69]. He introduced the doubling-halving algorithm, which describes the frequency architecture of EEG rhythms during conscious cognition, which are coupled to body oscillations such as heart rate and breathing with breathing centered at harmonic frequencies of 0.08 Hz (half of 0.16 Hz), 0.16 Hz and 0.32 Hz.

Different patterns of cardio-respiratory coupling can be discriminated: three subjects (#18, #3, #11, further referred to as group 1) displayed coherent frequency components in both signals in the LF band with a predominance of nRSA. Six subjects (#24, #16, #9, #13, #14, #20, further referred to as group 2) displayed a dominant respiration with a spectral peak close to 0.32 Hz, while RRI oscillations in the LF band prevailed. These subjects presented rate ratios of 1:n between RRI and respiratory rhythms with n > 1. The preferred rate ratios were 1:2 (2 breaths per RRI wave at an RRI rate of ~0.16 Hz) and 1:3 (3 breaths per RRI wave at an RRI rate of ~0.1 Hz). Only in subject #24, fast RRI waves were most prominent, but a small spectral peak at 0.1 Hz indicated that these fast RRI waves were superimposed on a weak slow RRI wave. In three subjects from group 2, nRSA was predominant; the other three subjects showed either predominant RSA or both RSA and nRSA to a similar extent. 

All these nine subjects from groups 1 and 2 exhibited “neural BOLD” oscillations, which are reflected in pTD between RRI waves and BOLD oscillations (i.e., RRI waves precede BOLD oscillations). An exception is subject #6 (further referred to as group 3) showing vascular BOLD oscillations, which are associated with nTD with RRI waves lagging behind the BOLD oscillations. 

Besides RSA and nRSA, many periods presented phase-relations neither belonging to RSA nor to nRSA (termed indefinite RSA; [32]). In cases of 1:1 coupling, wave-by-wave analysis allows a reliable assessment of RSA/nRSA. However, in cases of 1:n coupling (with n > 1), a large number of breaths is not considered in the analysis, thus giving rise to some uncertainty in the assessment of the percentage of RSA/nRSA. This applies to 8 cases presented in Table 1 (all except for subjects #11 and #18).

### 3.1. BOLD Oscillations of Neural and Non-Neural Origin

Three different signals are related to brainstem functions, namely (i) the fluctuations of cardiac beat-to-beat interval controlled by the cardiovascular center, (ii) the respiratory rhythm associated with the pre-Bötzinger complex modulated by cortical afferences, and (iii) BOLD signals with neural and non-neural origins. 

Neural BOLD oscillations are associated with rhythmically activated neural clusters with a pTD of ~2–3 s. In contrast, non-neural BOLD oscillations can be either vascular BOLD oscillations with origin in the baroreflex loop (subject #6) or respiratory artefacts caused by vessel motion. Examples of 3 subjects from group 1, all with neural BOLD oscillations, are presented in Figure 2. The figure shows averaged waves of 12 s duration (±SEM) composed of about ~10 single visually selected trials aligned to the maximum of the RR interval (RRI peak). The displayed waves are BOLD, breathing and RRI signals. All BOLD waves in line A are from the left precentral gyrus (ROI 1), the BOLD waves in line B are from the left brainstem (ROIs 103, 105), and BOLD waves in line C are from the right brainstem (ROIs 96, 100); for further details, see [33]. Breathing waves are coherent with RRI waves in the case of nRSA (subjects #11 and #18) and in opposite phase for RSA (subject #3). Note, neural BOLD waves lag behind the respiratory BOLD artefact or the RRI peak by about 2–3 s.

### 3.2. Cardiac RRI, Respiratory and BOLD Signals in the 0.05–0.15/0.1–0.2 Hz Band

The subjects of group 1 (#11, #18, #3) presented RRI and respiratory signals with dominance in two frequency bands centered at 0.1 Hz and at 0.16 Hz, which are assigned to the 0.05–0.15/0.1–0.2 Hz band. While BOLD waves displayed similar patterns across all subjects (examples are illustrated in Figure 2), differences were observed in RRI and breathing waves. Two subjects from group 1 (#11, #18) predominantly exhibited an in-phase (coherent) behavior of RRI and breathing waves while the other subject (#3) also showed an out-of-phase behavior over parts of the record. The latter case means that inspiration is accompanied by acceleration of the cardiac activity and expiration is associated with deceleration. This is characteristic for a positive respiratory sinus arrhythmia (pRSA or, shortly, RSA; see Figure 1). The former cases demonstrating that inspiration is accompanied by RRI increase (cardiac deceleration) and expiration by RRI decrease are characteristic for a negative RSA (nRSA; see Figure 1). The important point is that both types of patterns (negative and positive RSA) displayed the same time shifts between BOLD signals from the right and left brainstem (see Figure 2, lines B and C) and between breathing and BOLD signals (see Figure 2, lines C and D). 

Besides RSA/nRSA, the wave-by-wave analysis of respiratory and RRI signals in the three frequency bands centered at 0.1 Hz, at 0.16 Hz and at 0.32 Hz provides further features of cardio-respiratory coupling, such as abrupt PD changes (termed “PD transitions”) [30]. Two typical examples from #11R1 and #18R1 are presented in Figure 3 and Figure 4. In both records, respiratory and RRI waves range in the 0.1–0.2 Hz band, and the phase relation between them indicates nRSA (Figure 3A,B and Figure 4A,B). They are associated with slow BOLD waves (PD about 6 s) in the PCG and left pons lagging behind the RRI waves, characteristic of neural BOLD oscillations. In the amygdala, waves in the different frequency bands constitute a non-sinusoidal wave resulting from a superposition of oscillations at ~0.3 Hz on a slow wave of ~0.1 Hz. The time delay between a respiratory BOLD artefact in the right pons induced by vessel (basilar artery) motion and the neural BOLD oscillations is 2–3 s corresponding to the neuro-BOLD coupling time (Figure 3B and Figure 4B). Changes in the RRI PD are often accompanied by corresponding changes in the respiratory PD. In the cases presented here (#11R1 and #18R1), the transitions occurred first in the RRI rhythm and were followed by a transition in the respiratory rhythm (periods marked in red in Figure 3A and Figure 4A), indicating that the RRI rhythm has entrained the respiratory rhythm, which is typical for nRSA. However, the analysis of directed information flow (Figure 3C and Figure 4C) revealed differences between the two subjects. In #11R1, the information flow from respiration to RRI was predominant and was stronger than the opposite flow from RRI to respiration. In contrast, in #18R1, the information flow direction from RRI to respiration was clearly predominant. These are two remarkable examples showing clearly that the phase relation only shows one way in the cardio-respiratory interactions while the analysis of directed information characterizes reciprocal connections and the time variation of flow, thus providing a more differentiated representation of these processes. 

### 3.3. Interactions of RRI, Respiratory and BOLD Signals in the 0.2–0.4 Hz Band

Group 2 of our subjects (#24, #16, #9, #13, #14, #20) presented signals in three frequency bands. While in the RRI signals, two periods were dominant (average PDs 10.2 s = 0.1 Hz and 6.7 s = 0.15 Hz), the respiratory signal predominated in the 0.2–0.4 Hz band (average PD 3.1 s = 0.32 Hz). All subjects of group 2 presented cardio-respiratory coupling with higher integer rate ratios, mainly 1:2 or 1:3, indicating that 2 or 3 breaths are coinciding with 1 RRI cycle (see Table 1). In those cases, we often observed superpositions in the slower rhythm (i.e., in the RRI rhythm) [30]. This phenomenon results when the strength of the dominant rhythm (usually, the respiratory rhythm) is not sufficient to fully entrain the other rhythm (the RRI rhythm). An example is shown in Figure 5 (#14R4). Figure 5A shows recordings from respiration and RRI waves. The vertical dotted lines show the coincidence of expiration onset with local maxima of RRI, a coherent pattern clearly indicating nRSA. The percentage of nRSA in this resting state was 58%. Frequency analysis demonstrated a clear 1:3 cardio-respiratory rate ratio with 3 breaths per one RRI cycle (Figure 5A,B). Figure 5B shows a movement artefact in the right pons coherent with respiration (lower line, left panel), slow BOLD waves in the left pons and MFG coherent with slow RRI waves (right panels) and a phase-shifted BOLD oscillation of neural origin in the amygdala (lower line, middle panel). The analysis of directed information flow confirmed this finding, showing a pronounced information flow from RRI to respiration (Figure 5C).

In contrast to the subjects from groups 1 and 2, subject #6 (group 3) showed BOLD peaks that preceded the RRI peaks by about 2 s, indicating that these BOLD oscillations were associated with dominant fluctuations in cerebral blood flow and volume (vascular BOLD) and had their origin in the baroreflex loop (Figure 6).

### 3.4. Dominant Information Flow between Cardiac (RRI), Respiratory and BOLD Oscillations in the 0.1–0.2 Hz and 0.2–0.4 Hz Bands

We found heterogenous patterns of information flow between cardiac (RRI), respiratory and BOLD oscillations among the 10 subjects with high anxiety in the fMRI scanner. In seven subjects (groups 2 and 3), respiration was dominant in the HF band (0.2–0.4 Hz), which corresponds to a “normal” respiratory rate at rest. In three subjects (group 1), respiration was dominant in the 0.1–0.2 Hz band, which is rather seldom during unconscious spontaneous breathing but is typically for conscious breathing such as resonance breathing [27]. It is noteworthy that all subjects except one (#6R1) displayed neural BOLD oscillations with RRI waves preceding BOLD waves (pTD). In addition, we observed, in most of the subjects, a clear predominance of information flow from the RRI to the respiratory rhythm, as it was expected during elevated anxiety. Coupling properties such as a predominant information flow from the RRI to the respiratory signal and neural BOLD oscillations with pTD to RRI waves are typically accompanied by a nRSA coupling pattern, i.e., a coherence between respiratory and RRI rhythms. 

Previous DTF studies in subjects with prevailing anxiety revealed a rhythmic neural source (pacemaker) in the brainstem operating in the frequency range 0.1–0.2 Hz as a driving force for RRI signals [48]. We wondered whether a similar information flow can be demonstrated using DTF analysis in the 0.2–0.4 Hz band, indicating that modulations of cardio-respiratory coupling observed in subjects with high anxiety are present across wide ranges of respiratory and RRI frequencies. The DTF data from the subcluster of 6 subjects (group 2) were averaged and statistically compared via a bootstrap approach (Figure 7). Characteristic for these subjects is a predominant respiration in the 0.2–0.4 Hz frequency band (fast respiration). Most prominent was a significant top-down information flow from the MFG to the brain stem, the RRI and the respiratory rhythm (Figure 7A,B). Of note, there is a small information flow from respiration to the cortical and brainstem ROIs as well as to RRI. In contrast, the information flow from RRI to respiration was significantly greater than that from respiration to RRI (*p* = 0.01). These results clearly indicate that in situations of elevated anxiety, a significant information flow from RRI to respiration is not only demonstrable in the LF and IMF bands but also in the HF band. It may account for the reversal of the cardio-respiratory entrainment with a reduced dominance of the respiratory rhythm over the RRI rhythm in situations of elevated anxiety.

### 3.5. Management of Anxiety

All participants analyzed in this study showed high anxiety at least in the first session in the scanner. While their average AS in session R1 was 24 ± 4.1, it decreased by session R4 to 16 ± 5.4. However, there were considerable differences in the success of reducing anxiety (Figure 8). While the subjects in group 1 reduced their AS from R1 to R4 by more than 50% (ranging from 48–64%; *p* = 0.016, paired t test), the change of AS was widely scattering in groups 2 and 3 (*p* = 0.20, paired t test): two subjects in group 2 (#20 and #24) reduced their AS clearly (by 36 and 60%, respectively). Of note, nRSA prevailed in these two participants (see Table 1). Subject #6 (group 3) showed a similar reduction in AS. However, in two subjects (#9 and #14), AS remained almost unchanged throughout the whole experiment, while in two other subjects (#13 and #16), AS even increased over the two-hour fMRI session. Across all 10 participants, there was no significant correlation between the percentage of nRSA and reduction of AS (r = 0.32, *p* = 0.37).

To summarize, the results show that cardio-respiratory phase-relations may indicate different entrainment processes between brain and body rhythms involved in regulation of anxiety. However, the percentage of RSA/nRSA as an indicator of anxiety or anxiety management contains some uncertainty, especially in the HF band. Our results show that the directed information flow allows a more comprehensive assessment of the interplay between brain and body rhythms in a wide frequency range. Slow breathing seems to be an important component of successful anxiety management, presumably via the activation of a central pacemaker in the brainstem operating in the 0.15–0.16 Hz range. At higher breathing rates, respiration-entrained oscillations in different brain areas indicate that more pacemakers operating at different frequencies may be involved in anxiety management.

## 4. Discussion

The interactions between respiratory, RRI and BOLD signals from different brain regions provide insights into the strategies of anxiety management. In particular, the direction of information flow indicates different types of interaction between the involved processes and brain regions. All subjects presented in this study were characterized by high anxiety in at least one of four sessions within a two-hour fMRI examination. The results indicate that these subjects used different strategies to manage their anxiety. Some of these subjects showed high AS values in the first session and a decrease of AS up to the last session. In four subjects, however, AS did not decrease or even increased throughout the sessions, indicating a less successful anxiety managing process. The main and most evident difference between the subjects was their breathing rate: While seven subjects (groups 2 and 3) breathed at “normal” breathing rates (0.32 Hz on average), three subjects (group 1) presented slow breathing with prevailing breathing rates in the 0.05–0.15 Hz band (see Table 1). The groups further differed by the predominant directions of information flow and their success in anxiety management.

### 4.1. Different Cardio-Respiratory Coupling during High Anxiety

#### 4.1.1. Slow Breathing during High Anxiety (Group 1)

Group 1 consisting of subjects #18, #3 and #11 showed RRI and respiratory oscillations in the LF/IMF band with a predominant 1:1 rate ratio. RRI and respiratory waves were either in-phase (nRSA) or out-of-phase (RSA), but the portion of nRSA was higher than that of RSA in all three subjects (see Table 1). All three subjects presented their highest AS values in session R1 and significantly reduced their AS by more than 50% on average up to session R4 (see Figure 8).

Noteworthy, nRSA is not associated with a change in magnitude of RSA. Cardiovascular or emotional disorders are usually associated with a reduction in the magnitude of RSA, i.e., reduced cardiorespiratory coupling [16,17,18,20,23]. Likewise, sympathetic activation induced by graded postural challenge diminished cardiorespiratory coupling [15]. In contrast, all patterns of respiration-related RRI oscillations observed in this and in previous studies showed no indication of reduced cardiovascular coupling [29,30,70].

nRSA was typically observed in subjects breathing at a lower rate than the “normal” resting breathing rate [29,31,32]. Besides slow respiration, the common feature of the three subjects in group 1 was coherent neural BOLD oscillations in the PFC and brainstem centered at ~0.16 Hz and ~0.1 Hz, respectively (Figure 3 and Figure 4), indicating activation of a pacemaker in the brainstem. The BOLD peaks were clearly delayed from the largest RRI interval (RRI peak) by 2–3 s characteristic for an associated neural BOLD component. An interesting counterexample is subject #6, which is discussed below. 

Figure 3 and Figure 4 exhibit characteristic examples of RRI peak-triggered averaged RRI, breathing and BOLD waves. Both figures have in common that they represent subjects with nRSA showing slow neural BOLD waves in the brainstem and PCG as well as slow BOLD waves with superimposed fast wavelets in the amygdala (ROI 41). However, there exists an interesting difference. Subject #18, on the one hand, presented a strong information flow from RRI to the respiratory signal (Figure 4C) and from the MFG to respiration and brainstem (not shown). On the other hand, in subject #11, the information flow from respiration to RRI was dominant (Figure 3C) and also the flow from brainstem to the PCG (not shown). It is speculated that in the former case (#18R1), nasal breathing was predominant, while in the latter case (#11R1), a second possible pathway was active. This second pathway is thought to originate from the preBötzinger complex and to project to the suprapontine nuclei (via the locus coeruleus in the pons and olfactory nuclei [71] as well as to the central medial thalamus, which is directly connected to the limbic and sensorimotor cortical areas [72]. The nasal route of breathing seems to have particular importance for breathing-entrained oscillations [37,73,74,75] as is discussed in more detail below. Further research is necessary to specifically study the importance of nasal breathing for the management of anxiety.

#### 4.1.2. Fast Breathing during High Anxiety (Groups 2 and 3)

Group 2 consists of six subjects, which presented respiratory oscillations in the HF band. In all except one (subject #24), RRI oscillations were predominant in the 0.1–0.2 Hz band. In four subjects of this group, AS values did not decrease or even increased from session R1 to session R4 (see Figure 8). 

All six subjects in group 2 displayed delayed slow BOLD oscillations in the left PFC and the left brainstem, similar to that observed in group 1. Typically, RRI oscillations and respiration were coupled at higher integer rate ratios such as 1:2 or 1:3 (for an example, see Figure 5). Despite these high-integer rate ratios, there was also tight phase-coupling. This is in accordance with observations from motor-respiratory coupling studies demonstrating that a near-integer rate ratio favors stable entrainment [76]. Only in three subjects, the portion of nRSA was clearly higher than the portion of RSA; in the others, RSA and nRSA had a similar incidence or RSA even prevailed (see Table 1).

The most interesting part in this group is the highly significant information flow from MFG to the brainstem, to respiration and to RRI signals in the 0.2–0.4 Hz band (see Figure 7). Such a directed flow is characteristic for brain rhythms in the 0.1–0.2 Hz band entrained by nasal respiration and was recently reported and discussed [48]. The present results demonstrate that respiratory entrainment of brain rhythms is not confined to the 0.1–0.2 Hz band but also exists in the 0.2–0.4 Hz band. 

Subject #6 (group 3) represents a particular case, as they were the only subject with high anxiety and vascular BOLD oscillations. This participant showed slow RRI and fast respiratory oscillations but no slow BOLD oscillations after slow RRI oscillations (Figure 6). The implications of this specific observation are discussed below. 

### 4.2. Coupling of Respiration with Heart and Brain Oscillatory Activity

#### 4.2.1. Phase-Shifts in Respiration-Related RRI Oscillations and Activation of a Central Pacemaker

The RRI increase is never an isolated process and is always accompanied by a simultaneous desynchronization of EEG alpha and/or beta rhythms. One example of the common RRI increase and EEG desynchronization is the Orienting Reflex [77,78], another the decision-making process [79]. Changes in EEG activity are associated with metabolic processes and a causal BOLD response [80]. As in the case of nRSA, the increase of RRIs coincides with inspiration (activation of the preBötzinger complex), the corresponding BOLD response is of large magnitude. Dominant slow BOLD oscillations in the brainstem, therefore, point more to nRSA than to RSA.

Among cardio-respiratory interactions, the respiratory modulation of the cardiac RRI (RSA) is certainly the most prominent one [5,11,39]. For classical RSA, it is widely accepted that the respiratory rhythm is the leading one entraining the fluctuations of RRI [2,17,30]. However, this is not the only phase-relation induced by cardio-respiratory coupling. A detailed analysis of cardio-respiratory phase-relations demonstrated an inverse phase-relation, termed negative RSA (nRSA) and many phase-relations summarized as indefinite RSA [32]. Such phase-shifts are thought to result from a change in the direction of entrainment between the cardiac and the respiratory rhythm. The wave-by-wave analysis is an appropriate instrument to characterize frequency and phase coupling [29]. It allows to differentiate changes in rate ratios, which may result from simple rate changes, from true modulations of coupling as may be indicated by shifts in the phase-relation between the coupled rhythms. These modulations can give hints on the dominance of the rhythms. Using a wave-by-wave analysis, we recently demonstrated that nRSA was mostly associated with a change in dominance, i.e., the RRI rhythm was the leading rhythm and entrained the respiratory rhythm [30]. This was confirmed by the present results of the DTF analysis revealing a significant information flow from the RRI to the respiratory rhythm (see Figure 7). However, only few of our subjects showed prevailing nRSA in the recorded sessions. Many subjects presented both RSA and nRSA at similar percentages (see Table 1), indicating that such a switching in dominance is a rather occasional event.

As nRSA is an untypical coupling pattern, it is unclear which factors may promote the change of dominance and the switching in the cardio-respiratory phase-relation from the common RSA pattern to the inverse nRSA pattern. The activation of an additional neural rhythm integrated in the cardio-respiratory coupling might account for the supposed change of dominance. Such phenomena have often been reported in studies on motor-respiratory coupling [76,81,82,83]. Usually, coupling between two or more concomitant rhythms is based on a mutual interaction. This means that the dominance can fluctuate, so that each of the concomitant rhythmic processes can take the lead for a limited interval of time. If three or more oscillators are coupled, synchronization between two oscillators (e.g., acoustic pacing of limb rhythms or synchronous activity of several muscle groups) increases their frequency- and phase-locking effects onto a third one (e.g., respiration) and stabilizes coupling [81,84,85,86]. In a previous study, we investigated PLV-derived time delays between slow RRI oscillations and slow BOLD signals in the frequency range of 0.10–0.15 Hz and showed that neural BOLD oscillations lagging behind the RRI waves indicate activation of a central pacemaker [33]. Such a pacemaker activity located in the reticular formation of the brainstem operating in the frequency range around 0.15 Hz has been proposed by Perlitz and co-workers [87]. This “0.15-Hz rhythm” that was observed both in men and dogs was synchronized with all cardiovascular-respiratory oscillations. Of note, this synchronization is not confined to 1:1 coupling but also includes higher integer rate ratios [87]. This one or a similar rhythm might be candidates reflecting pacemaker-like processes activated during emotional regulation and anxiety management, which may account for a switching in dominance. The assumption that activation of a central pacemaker-like mechanism is involved in the management of scanner-related anxiety is supported by stable phase-coupling between respiratory and RRI oscillations at 1:1 or at higher integer rate ratios and by the occurrence of slow neural BOLD oscillations in nine of ten subjects of the present study. Another support, in the reverse direction, comes from subject #6 presenting slow RRI oscillations and vascular BOLD oscillations, but no slow neural BOLD oscillations. If no pacemaker in the brainstem is activated, no associated BOLD oscillations can be expected in the brainstem or in the cortex. Moreover, a DTF analysis recently revealed a significant information flow in the 0.1–0.2 Hz band from the pons to the precentral cortex and to the RRI signal. This flow was interpreted to result from the activation of a rhythmic neural source in the brain stem operating at about 0.15 Hz (such as Perlitz’ “0.15-Hz rhythm”) [48]. However, the nature of the hypothesized pacemaker or pacemakers cannot be identified from the present data. In particular, neural processes involved in autonomic regulation would be candidates of interest. A recent study on causal relations among heart period, systolic arterial pressure and respiration and their interactions with autonomic control showed that bidirectional interactions between cardiac and respiratory rhythms were not modified by parasympathetic or sympathetic blockade [88]. 

#### 4.2.2. Respiration-Entrained Brain Oscillations and Possible Strategies of Anxiety Management

The respiratory rhythm provides a continuous rhythmic modulation of cortical neuronal activity that, in turn, modulates autonomic, sensorimotor, cognitive and emotional processes [17,84,89,90]. Respiration-entrained oscillations in neural activity are ubiquitous in the brain. They have mostly been described in rodents, e.g., in the parietal and in the frontal cortex [91,92], but also in large areas of the human cortex [75,90,93]. Of note, most of these observations in humans were mostly made during nasal breathing, indicating a particular role of the olfactory tract [37,73,74,75]. Nasal breathing induces rhythmical bursting of olfactory receptor neurons, which in turn entrains activity of the olfactory bulb and the piriform cortex [75]. Moreover, respiration can entrain the activity of various cortical and subcortical structures such as the prefrontal cortex, amygdala and hippocampus [37,91,93], thus forming a wide-spread network of respiration-modulated brain oscillation across all major frequency bands [73]. Nasal respiration significantly modulated cortical excitability, and these changes were followed by respiration-locked changes in perceptual sensitivity [74]. A comparison of oral and nasal breathing revealed a significant influence of the nasal but not the oral respiratory phase on the performance of emotion and memory tasks, which were closely related to the function of the amygdala and hippocampus [37], thus suggesting a role of the nasal route of breathing in emotional regulation.

As healthy awake subjects predominantly breathe through their nose, it is likely that nasal breathing might also be preferred in the subjects of the present study. We would therefore assume that respiration-entrained oscillations in cortical and limbic areas are involved in the strategies of anxiety management. Upward flows from respiration to cortical and pontine ROIs as well as to RRI have been observed in the present study even though they seemed unimportant in comparison to the highly predominant downward flows from MFG to the brainstem and to RRI and respiration (see Figure 7). We would speculate that one or more pacemakers operating in a frequency range centered at ~0.10 Hz and/or ~0.16 Hz are activated through nasal breathing. These pacemakers may primarily entrain the RRI rhythm, and the coupled rhythms then entrain the rhythm generated by the medullary respiratory neurons. This means a reduced dominance of the respiratory rhythm over the RRI rhythm that favors a shift in the phase-relation towards nRSA. Alternatively, the pacemaker rhythm primarily entrains the medullary respiratory rhythm, which subsequently entrains rhythms in other parts of the brain such as in the middle frontal cortex and in the brainstem. In this case, respiration also entrains the RRI rhythm, which may stabilize (classical) RSA. The results suggest that activation of one or more pacemakers operating especially in the 0.1–0.2 Hz band may be an important mechanism of managing high anxiety. The particular case of subject 6 presenting no neural BOLD waves indicates that activation of such a pacemaker is not the only way of successful anxiety processing.

In turn, the reduction of anxiety over the total fMRI experiment varied widely among the 9 subjects with neural BOLD oscillations where activation of a pacemaker can be assumed. Another important mechanism for successfully managing anxiety is slow breathing as can be concluded from the comparison between groups 1 and 2. Since hundreds of years, slow breathing has been used in various human societies, particularly in eastern cultures, for strategies of calming down, relaxing and emotional processing such as in yoga, qigong or other types of meditation (for a review, see [94]). Resonance breathing at about 6 breaths/min as used in biofeedback training has been reported to increase the amplitude of heart rate variability, which is considered to contribute to stress reduction and emotional regulation [26,27]. Nasal breathing is an essential component of such breathing techniques [95,96].

### 4.3. Limitations and Future Directions

This study has several limitations. A major limitation is that measurement of respiration did not allow to differentiate between nasal and oral breathing. The nasal route of breathing, especially the olfactory tract, seems to have particular importance for entrainment of brain rhythms to respiration [37,73,74,75]. We have good reason to suppose that our subjects mainly performed nasal breathing as healthy awake subjects prefer to breathe through their nose, but we cannot estimate whether and to which extent oral breathing may have contributed to their breathing pattern. For accurate assessment of the proposed mechanisms, a comparative study with nasal and oral breathing is necessary. 

A further limitation of the study is that assessment of anxiety was only based on questionnaires for AS values. This was mainly due to the original intention of the fMRI study. At the time of planning the experiment, the main interest was directed on the initiation of self-paced movements and their relation to 0.1 Hz brain oscillations. The question of scanner-related anxiety came into focus during the experiments and the evaluation period. Therefore, no further psychophysiological measure of anxiety such as the skin conductance response was included.

The original intention of the fMRI experiment as described above may also account for the lack of collecting more data, reflecting autonomic regulation and autonomic functions. For instance, examination of further respiratory parameters such as tidal volume, ventilation or end-tidal partial pressure of CO_2_ might indicate a possible hyperventilation, which, in turn, affects CBF. Continuous blood pressure measurements would provide more information on the relation of the baroreflex pathway with the RSA patterns observed in this study. In addition, those parameters might corroborate the information obtained by the AS values. 

Many chronic cardiovascular diseases are associated with affection of HRV, in particular, with a change in HRV magnitude even up to abolishment of RSA [7,16,17]. Reduced HRV magnitude is also characteristic for emotional dysregulation [20,23] and for sympathetic activation [15]. In the present study, we focused on the phase-relationship between respiratory and RRI rhythms and did not explicitly analyze the amplitude of respiration-related RRI oscillations. This is a limitation as the amplitude of HRV may give additional information, e.g., on the sympathetic or vagal activation in the processing of anxiety. Future studies on interactions between heart, respiration and brain signals should include analyses of HRV magnitude.

Previously, we have applied a wave-by-wave analysis to characterize the coupling between respiration and RRI oscillations. Using this method, we were able to exactly assess the phase-relations between these rhythms and to identify nRSA, preferably in subjects with slow breathing and slow RRI waves [29]. The specific cardio-respiratory phase-relations (RSA or nRSA) may be masked when the respiratory rate considerably differs from the rate of RRI oscillations as reflected in a predominance of 1:2, 1:3 or even higher cardio-respiratory rate ratios. The main problem with high breathing rates and higher-integer cardio-respiratory rate ratios is that the number of cardiac periods per breath decreases at higher breathing rates, thus complicating the assessment of RRI changes during inspiration or expiration. In addition, only every second or third breath is included in the analysis of phase-relations between respiratory and RRI oscillations. As a result, the number of evaluated periods was reduced, and the percentage of indefinite RSA increased while the portion of positive or negative RSA types decreased.

The wave-by-wave analysis allows a precise and reliable assessment of cardio-respiratory phase relations if (a) the period durations (PD) of RRI waves and breathing cycles are similar, i.e., the rate ratio between RRI and breathing waves is about 1:1, and (b) both rhythms are slow (below ~0.2 Hz), that means, in the low-frequency (LF) and intermediate-frequency (IMF) bands. With respiratory and/or RRI waves at higher rates, detection of nRSA according to wave-by-wave analysis contains a degree of uncertainty and has only limited significance. The results of this study demonstrated that the combination with causal coupling analysis outweighs this disadvantage and provides additional and more detailed information, in particular in the HF band. The slight discrepancies between the results of the two analysis methods can easily be explained by the different meanings of the results: The directed information flow characterizes the cause while the phase-relationship characterizes the consequence of the modulation in the coupling between cardio-respiratory and brain rhythms.

## 5. Conclusions

Previous studies on participants of fMRI examinations showed that persons with elevated anxiety presented a rather unusual cardio-respiratory coupling pattern characterized by an inverse phase-relation (compared to the normal RSA), termed as nRSA [29,70]. We had hypothesized that a “central pacemaker”, which is involved in emotional regulation and activated in situations of elevated anxiety, would entrain the RRI rhythm and thus modulate cardio-respiratory coupling [30]. The present results revealed a prevalence of neural BOLD oscillations in the brain stem, supporting this assumption. In addition, the DTF analyses revealed a strong directed information flow from the RRI rhythm to the respiratory rhythm, thus providing additional support to this hypothesis. Moreover, DTF analysis data indicated that respiration-entrained brain oscillations are involved in strategies of anxiety regulation. However, further research is necessary to prove this assumption. 

## Figures and Tables

**Figure 1 biomedicines-11-01028-f001:**
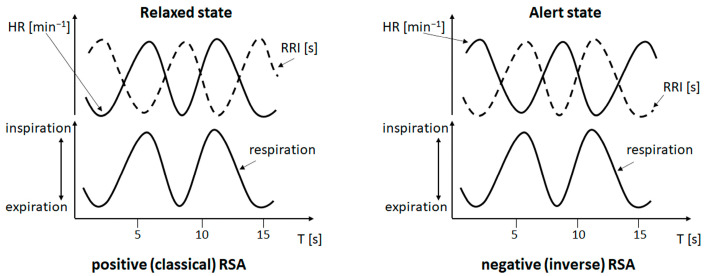
Cardio-respiratory phase-relations: upper curves show heart rate (HR, solid line) and cardiac beat-to-beat interval (RRI, dotted line), lower curves show respiration. Slow cardiac and respiratory waves are presented in both panels for reasons of comparability. The left panel depicts positive (classical) RSA with inspiration coinciding with increasing HR (decreasing RRI) and expiration coinciding with decreasing HR (increasing RRI). This phase-relation typically occurs in a resting and relaxed state. The right panel depicts negative (inverse) RSA with inspiration coinciding with decreasing HR (increasing RRI) and expiration coinciding with increasing HR (decreasing RRI). This phase-relation occurs rarely. In awake subjects, it has predominantly been observed in situations of elevated anxiety and during slow breathing.

**Figure 2 biomedicines-11-01028-f002:**
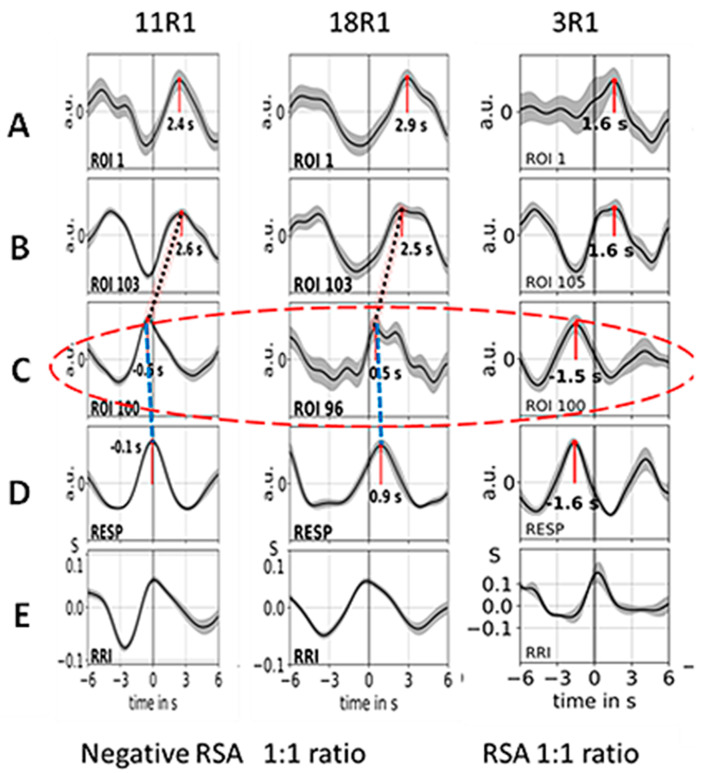
Examples of ~10 averaged BOLD (**A**–**C**), respiration (**D**) and RRI (**E**)-peak-triggered waves (mean ± SEM) of the subjects of group 1 (#11, #18, #3). Time intervals are shown from 6 s before to 6 s after the trigger. Numbers indicate the time shift in seconds between the peak and trigger. The red ellipse marks respiratory motion artefacts of the BOLD signal. The time shifts between breathing (line D) and respiratory BOLD artefacts (ROIs 96, 100; line C) are indicated by blue stippled lines, and the time shifts between respiratory BOLD artefacts and associated neural BOLD waves (ROI 103, 105; line B) are indicated by red stippled lines (only in the first two columns). The coherent breathing and RRI waves (lines D and E) in the first two subjects (#11, #18) indicate negative RSA (nRSA) while the out-of-phase behavior of breathing and RRI waves in the third column (subject #3) is characteristic for classical (positive) RSA (modified from [70]).

**Figure 3 biomedicines-11-01028-f003:**
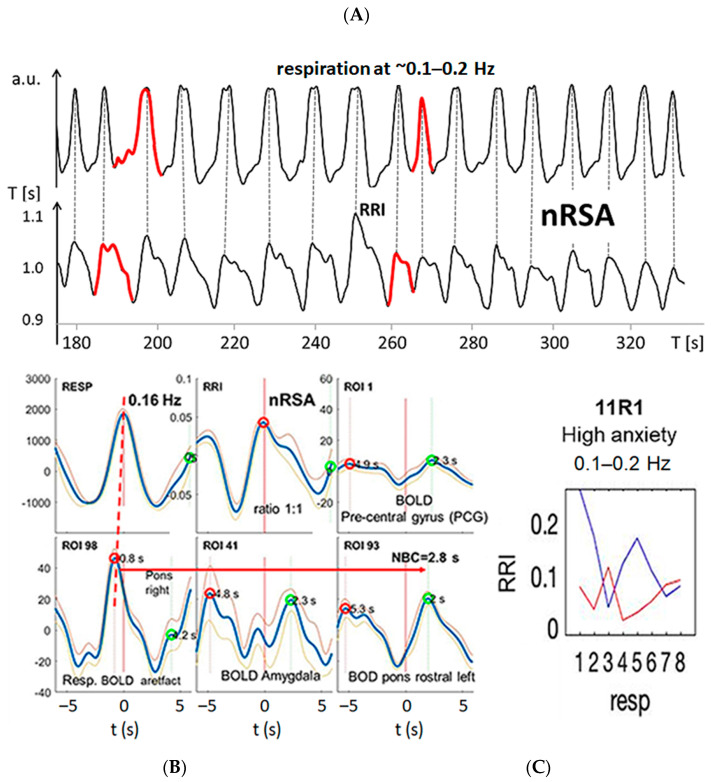
Example from subject #11. (**A**) Records of respiratory (upper trace) and RRI waves (lower trace); abscissa: time [s]. Some respiratory and RRI periods with PD transitions are marked in red. Note that transitions in the RRI rhythm precede changes in the respiratory rhythm by one period. (**B**) RRI peak (longest RR interval)-triggered respiratory (resp), RRI and BOLD waves from the PCG, pons and amygdala (blue and gray lines: mean ± SEM). The red and green circles have been automatically inserted by the computer program and mark local maxima on the negative and positive time scale, respectively. The coherence of respiratory and RRI waves indicates nRSA (upper line, left and middle panels). The vascular BOLD artefact in the right pons (lower line, left panel) slightly precedes respiration (red stippled line). The red horizontal arrow indicates the neural-BOLD coupling time (NBC) between the vessel motion artefact and the neural BOLD component. Note the non-sinusoidal oscillation in the amygdala (lower line, middle panel). (**C**) Information flow between RRI and respiration, blue lines indicate flow from respiration to RRI, red lines indicate flow from RRI to respiration. The numbers 1–8 on the abscissas represent consecutive time windows of the recording.

**Figure 4 biomedicines-11-01028-f004:**
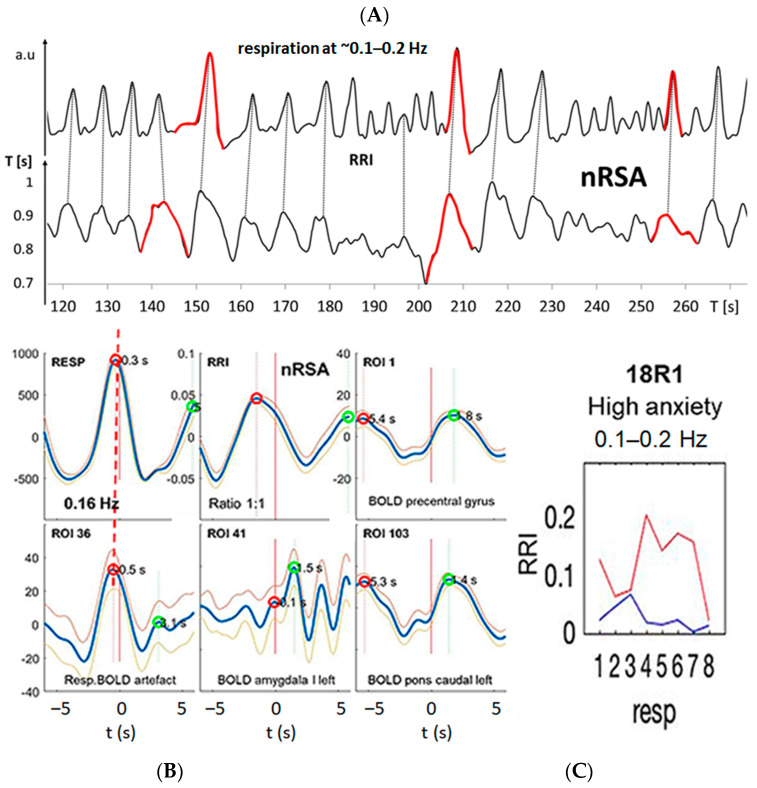
Example from subject #18. (**A**) Records of respiratory (upper trace) and RRI waves (lower trace); abscissa: time [s]. Some respiratory and RRI periods with PD transitions are marked in red. Note that transitions in the RRI rhythm precede the changes in the respiratory rhythm. (**B**) RRI peak (longest RR interval)-triggered respiratory (resp), RRI and slow BOLD waves (mean ± SEM; more details see text). Note the fast oscillation in the amygdala (lower line, middle panel). (**C**) Information flow is predominant from RRI to respiration; for further explanations, see Figure 3.

**Figure 5 biomedicines-11-01028-f005:**
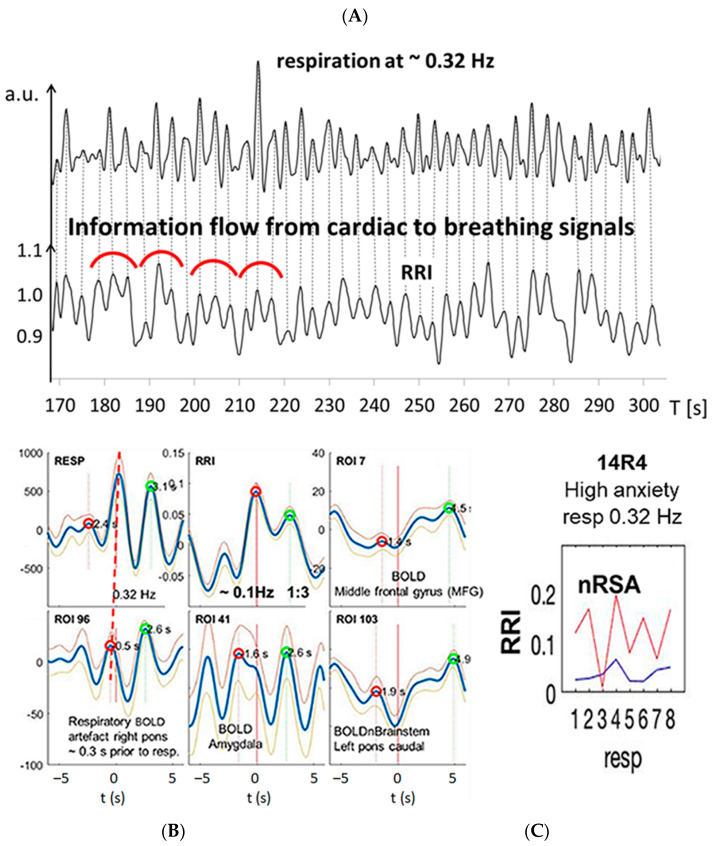
Example from subject #14 with dominant fast respiration in the 0.2–0.4 Hz band. (**A**) Records of respiratory and RRI waves. Some RRI periods with 1:3 coupling are marked with red semi-circles, indicating superposition of fast wavelets at ~0.3 Hz originating from the respiratory rhythm on slow RRI waves at ~0.1 Hz. (**B**) RRI peak (longest RR interval)-triggered respiratory (resp), RRI and slow BOLD waves (mean ± SEM; more details, see text). (**C**) The information flow diagram indicates a predominant flow from RRI to respiration. For further explanations, see Figure 3.

**Figure 6 biomedicines-11-01028-f006:**
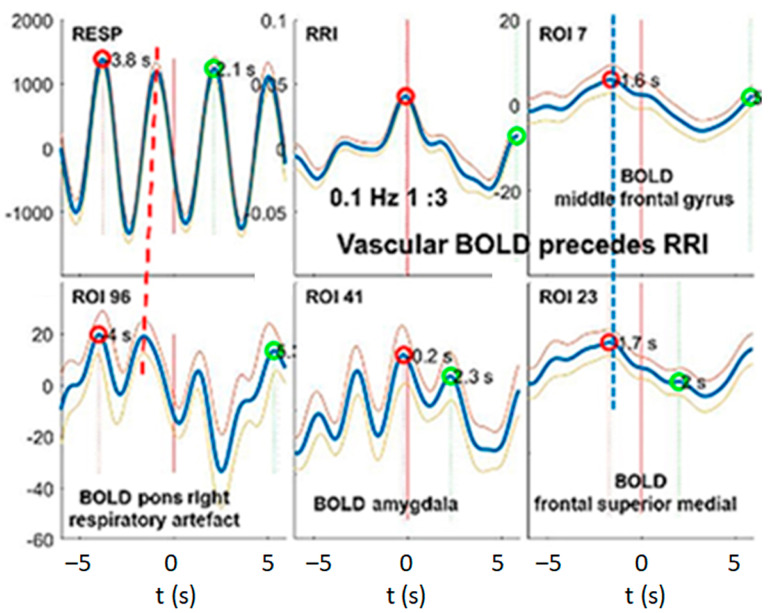
Example from subject #6 with dominant fast respiration in the 0.2–0.4 Hz band (for more detailed explanation, please see Figure 3). The RRI peak-triggered vascular BOLD waves were derived from MFG as depicted in the upper middle and right panels. The blue stippled vertical line indicates the maxima of slow BOLD waves about ~2 s before the RRI peak (longest RR interval), this means, a negative time delay indicating the vascular origin of the BOLD waves. Note the ~0.32 Hz oscillations in the amygdala (lower middle panel).

**Figure 7 biomedicines-11-01028-f007:**
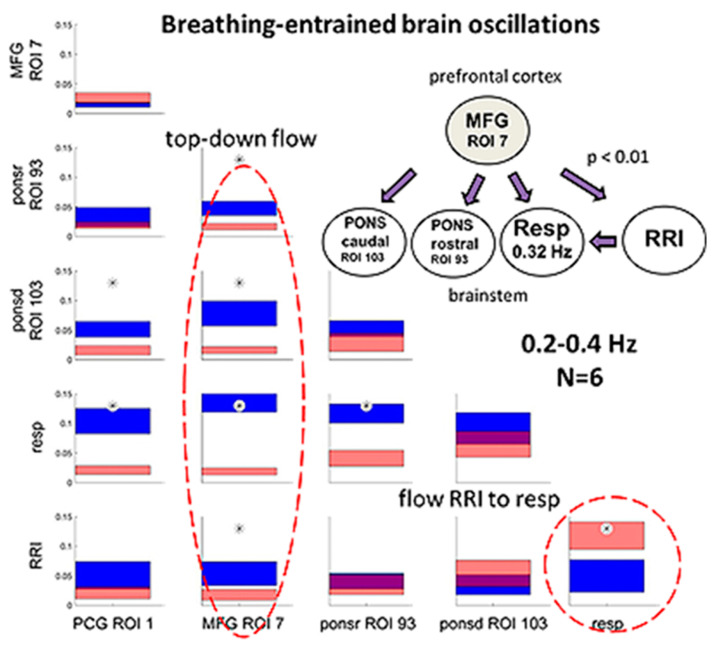
Directed coupling strengths for group 2 (n = 6) with high anxiety and predominant respiration in the 0.2–0.4 Hz frequency band. Each box shows the strength of coupling on the vertical scale. The width of bars is proportional to the mean error. The blue color shows the flow from the signal marked below the given column to the signal marked on the left, and the red color shows the flow from the signal marked on the left to the signal marked below. Violet areas mark an overlap of the two flow directions. Significant differences between couplings of inflow and outflow are marked by asterisks (*p* < 0.05). Red stippled ellipses mark the dominant downward flow (on the left) and the information flow from RRI to respiratory signal (on the right). The insert in the right upper corner indicates a diagram with the most important downward flows. Less pronounced flows, especially upward flows, are not shown.

**Figure 8 biomedicines-11-01028-f008:**
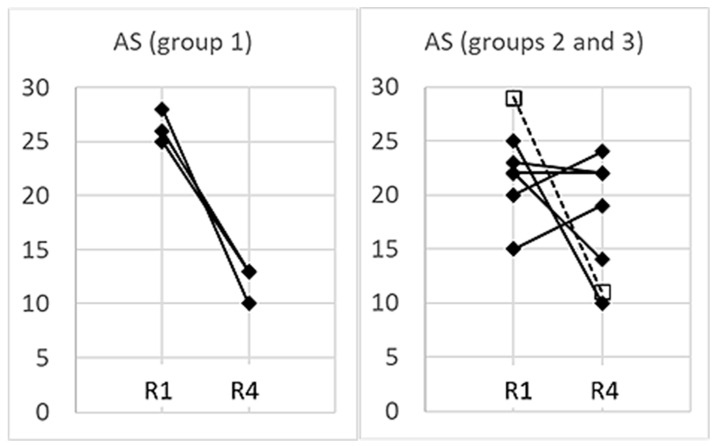
AS values obtained from all 10 subjects in sessions R1 and R4. In the interest of greater clarity, the subjects of group 1 are presented in the left panel and the subjects of groups 2 and 3 in the right panel. Subject #6 (group 3) is marked as ☐- - -☐.

**Table 1 biomedicines-11-01028-t001:** Results of the wave-by-wave analysis and BOLD analysis from 10 fMRI participants (allocated to 3 groups as described in the text) with high anxiety (modified from [30]).

Subject	AS	RSA/nRSA in % of RRI Waves		Average Rate		BOLD Analysis			RRI				Respiration					
									0.10 Hz Band		0.16 Hz Band		0.10 Hz Band		0.16 Hz Band		0.32 Hz Band	
		RSA%	nRSA%	RRI [Hz]	Resp. [Hz]	TD [s]	Sigbin%	ROI	PD [s]	n [%]	PD [s]	n [%]	PD [s]	n [%]	PD [s]	n [%]	PD [s]	n [%]
**Group 1**																		
#18R1	28	12%	63%	0.12	0.16	2.2	36%	103	9.8	47%	6.9	40%	10.0	22%	6.5	39%	3.2	26%
#3R1	26	13%	29%	0.21	0.22	1.7	24%	93	10.5	4%	6.5	13%	---	0%	6.4	15%	3.6	39%
#11R1	25	5%	80%	0.13	0.13	2.3	70%	93	9.8	39%	6.9	54%	9.9	38%	6.9	48%	---	0%
**Mean**	**26.3**	**10.0%**	**57.3%**	**0.15**	**0.17**	**2.1**	**43.3%**		**10.0**	**30%**	**6.8**	**35%**	**9.9**	**20%**	**6.6**	**34%**	**3.4**	**22%**
SD	1.5	4.4%	26.0%	0.05	0.04	0.3	23.9%		0.4	22%	0.2	21%	0.1	19%	0.3	17%	0.3	20%
**Group 2**																		
#24R1	25	12%	54%	0.25	0.26	1.5	26%	93	---	0%	6.8	2%	---	0%	5.9	6%	3.4	63%
#16R4	24	18%	14%	0.22	0.33	0.8	31%	93	9.9	11%	6.4	19%	---	0%	5.9	1%	3.0	97%
#9R1	23	58%	33%	0.14	0.33	1.9	36%	93	9.9	38%	7.0	33%	---	0%	---	0%	3.0	78%
#13R2	22	47%	41%	0.10	0.35	2.0	34%	103	11.7	61%	6.9	33%	---	0%	---	0%	2.8	87%
#14R4	22	6%	58%	0.21	0.31	1.4	27%	93	9.9	20%	6.5	10%	---	0%	---	0%	3.2	88%
#20R1	22	27%	47%	0.14	0.27	2.4	47%	93	9.6	42%	6.7	24%	---	0%	6.0	2%	3.4	65%
**Mean**	**23.0**	**28.0%**	**41.2%**	**0.18**	**0.31**	**1.7**	**33.5%**		**10.2**	**29%**	**6.7**	**21%**	**---**	**0%**	**5.9**	**1%**	**3.1**	**80%**
SD	1.3	20.5%	16.0%	0.06	0.04	0.6	7.7%		0.8	22%	0.3	13%	---	0%	0.1	2%	0.2	14%
**Group 3**																		
#6R1	29	11%	20%	0.17	0.34	−0.5	23%	97	10.3	29%	6.7	14%	---	0%	---	0%	3.0	93%

The columns denote (from left to right): subject; anxiety score (AS); percentage of RRI waves with RSA (RSA%) and nRSA (nRSA%), respectively; average rates of RRI waves and respiration (in Hz); results of phase-coupling between BOLD oscillations in the brainstem and RRI oscillations (time delay [s] (TD), significant length of phase coupling [%] (sigbin%), ROI with the largest time delay); period duration (PD) and percentage (n) of RRI waves in the frequency bands centered at 0.10 Hz and 0.16 Hz; PD and percentage of breaths in the frequency bands centered at 0.10 Hz, 0.16 Hz, and 0.32 Hz.

## Data Availability

Not applicable.

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
