# Peer review of "Analysis of Respiratory Sinus Arrhythmia and Directed Information Flow between Brain and Body Indicate Different Management Strategies of fMRI-Related Anxiety"

_biomedicines, 2023, doi:10.3390/biomedicines11041028_

Round 1

Reviewer 1 Report

Rassler and colleagues in the present article entitled ‘Patterns of respiratory sinus arrhythmia and the information flow between brain and body may provide information on management of elevated anxiety’, showed how subjects with high fMRI-induced anxiety use different strategies to manage their anxiety. Analyses of directed information flow and of cardio-respiratory phase-relations provided a clear insight into the coupling strategies: in these terms, analysis of phase-relations and of the occurrence of RSA/nRSA allows consistent inferences on the direction of coupling if the concomitant rhythms are coupled at a ratio of 1:1. At higher integer rate ratios (1:2 or 1:3 or even higher), allocation of cardio-respiratory phase-relations to RSA or nRSA becomes more and more uncertain, but DTF analysis still provides reliable information. Finally, authors concluded by confirming their assumption of a neural pacemaker activated in situations of high anxiety. 

In general, I think the idea of this article is really interesting and the authors’ fascinating observations on this timely topic may be of interest to the readers of Biomedicines. However, some comments, as well as some crucial evidence that should be included to support the author’s argumentation, needed to be addressed to improve the quality of the manuscript, its adequacy, and its readability prior to the publication in the present form. My overall judgment is to publish this paper after the authors have carefully considered my suggestions below, in particular reshaping parts of the ‘Introduction’ and ‘Methods’ sections by adding more evidence.

 Please consider the following comments:

A graphical abstract that will visually summarize the main findings of the manuscript is highly recommended.

Abstract: In my opinion, a lack of explanation of sinus arrhythmia definition and how it is mostly impacts on heart-rate variability components makes the reader unable to grasp the key concept of this article by looking at the abstract. I would suggest to reorganize this section, also without using abbreviations.

In general, I recommend authors to use more references to back their claims, especially in the Introduction of this meta-analysis, which I believe is lacking. Thus, I recommend the authors to attempt to expand the topic of their article, as the bibliography is too concise. Nevertheless, I believe that less than 60/70 articles are too low for a research article. Therefore, I suggest the authors to focus their efforts on researching relevant literature: in my opinion, adding more citations will help to provide better and more accurate background to this study. 

Introduction: The ‘Introduction’ section is well-written and nicely presented, with a good balance of descriptive text and information about cardio-respiratory coupling pattern in persons with elevated anxiety, termed as an nRSA. Still I believe that more information about effects of psychological stress on autonomic responses, including studies that focused on heart rate reactivity (HRV) and cardiac vagal reactivity (as indexed by respiratory sinus arrhythmia reactivity) and described how HRV has been employed as quantitative index of the interplay between sympathetic and parasympathetic influences on cardiac activity (https://doi.org/10.1016/j.tins.2022.04.003; https://doi.org/10.1111/psyp.14122). 

Participants: In my opinion, data about participants and information about clinical assessment for patients’ selection are not adequately explained. For this reason, I would ask the authors to specify inclusion criteria for patients involved in this study, like severity of disorder. Also, could the authors specify how did they estimate the exact number of participants and provide more information about the diagnostic tests used for clinical evaluation?

I was wondering, did the authors take into account the possibility combine the electrocardiogram (ECG) signal recording with an established psychophysiological measure of anxiety, that is the SCR?

I would ask the authors to include a proper and defined ‘Limitations and future directions’ section before the end of the manuscript, in which authors can describe in detail and report all the technical issues brought to the surface.

Figures: I suggest to modify all figures for clarity because, as it stands, the readers may have difficulty comprehending it. Also, please change the scale of the vertical axis and use the same minimum/maximum scale value in all the graphs.

I hope that, after these careful revisions, this paper can meet the Journal’s high standards for publication. 

I am available for a new round of revision of this article. 

Best regards,

Reviewer

Author Response

Responses to reviewer #1:

We thank this reviewer for his/her valuable and constructive comments.

  • A graphical abstract that will visually summarize the main findings of the manuscript is highly recommended.

Response: We have created a graphical abstract.

  • Abstract: In my opinion, a lack of explanation of sinus arrhythmia definition and how it is mostly impacts on heart-rate variability components makes the reader unable to grasp the key concept of this article by looking at the abstract. I would suggest to reorganize this section, also without using abbreviations.

Response: We have re-written the Abstract including a definition of respiratory sinus arrhythmia. The word count is now below 200 words without using abbreviations according to the journal’s guidelines.

  • In general, I recommend authors to use more references to back their claims, especially in the Introduction of this meta-analysis, which I believe is lacking. Thus, I recommend the authors to attempt to expand the topic of their article, as the bibliography is too concise. Nevertheless, I believe that less than 60/70 articles are too low for a research article. Therefore, I suggest the authors to focus their efforts on researching relevant literature: in my opinion, adding more citations will help to provide better and more accurate background to this study. 

Response: We have extended the Introduction section in order to better explain the background and the claims of this study. In particular, we have included more references into this section.

  • Introduction: The ‘Introduction’ section is well-written and nicely presented, with a good balance of descriptive text and information about cardio-respiratory coupling pattern in persons with elevated anxiety, termed as an nRSA. Still I believe that more information about effects of psychological stress on autonomic responses, including studies that focused on heart rate reactivity (HRV) and cardiac vagal reactivity (as indexed by respiratory sinus arrhythmia reactivity) and described how HRV has been employed as quantitative index of the interplay between sympathetic and parasympathetic influences on cardiac activity (https://doi.org/10.1016/j.tins.2022.04.003; https://doi.org/10.1111/psyp.14122). 

Response: We have re-organized and re-written the Introduction section and provided more information on the relation between HRV and autonomic balance. We thank this reviewer for his/her recommendations of literature, which have been included into the revised version.

  • Participants: In my opinion, data about participants and information about clinical assessment for patients’ selection are not adequately explained. For this reason, I would ask the authors to specify inclusion criteria for patients involved in this study, like severity of disorder. Also, could the authors specify how did they estimate the exact number of participants and provide more information about the diagnostic tests used for clinical evaluation?

Response: The participants of this study were healthy persons without reported disorders. We apologize that this has not become fully clear from the original manuscript. We have inserted some additional information into the Methods section characterizing the subjects and describing the selection criteria in more detail.

  • I was wondering, did the authors take into account the possibility combine the electrocardiogram (ECG) signal recording with an established psychophysiological measure of anxiety, that is the SCR?

Response: No, we did not. The main reason was in the original intention of the fMRI study. At the time of planning the experiment, the main interest was directed on the initiation of self-paced movements and their relation to 0.1 Hz brain oscillations. The question of scanner-related anxiety came into focus during the experiments and the evaluation period. Therefore, no further psychophysiological measure of anxiety such as the skin conductance response was included. We have added a Limitations section to the manuscript where these reasons are explained.

  • I would ask the authors to include a proper and defined ‘Limitations and future directions’ section before the end of the manuscript, in which authors can describe in detail and report all the technical issues brought to the surface.

Response: We thank this reviewer for this constructive proposal. As mentioned above in our previous response, we have created a section “Limitations and future prospects”. 

  • Figures: I suggest to modify all figures for clarity because, as it stands, the readers may have difficulty comprehending it. Also, please change the scale of the vertical axis and use the same minimum/maximum scale value in all the graphs.

Response: We have to apologize for the complexity of some Figures. These Figures should help the readers to understand the text better. Figs  3-5 include each 3 diagrams (graphs) with different signals: (A) epochs of RR interval time course and respiration ( B) averaged (mean+-standard error)  responses of respiration, RRI and BOLD signals  and (C) information about directed coupling between RRI and respiration signals for 8 time points. The vertical axes of the graphs in part B (all 3 Figures) indicate not only different signal qualities but also different scales with different maximal/minimal values. A harmonization of the scales of the vertical axis is therefore not possible.

Reviewer 2 Report

The study investigates the greater likelihood of finding negative respiratory sinus arrhythmia (nRSA) pattern during elevated anxiety via the acquisition of blood oxygenation level dependent (BOLD) signals from various regions of interest of the brain.

 The contribution is interesting and rich of physiological considerations. However, some issues need a more careful discussion that must account more profoundly the contributions present in literature.

1)      Cardiorespiratory coupling depends on the autonomic nervous system state. Sympathetic activation induced by graded postural challenge reduces cardiorespiratory coupling (see A Porta et al, Comput Biol Med, 42, 298-305, 2012). As a consequence of the reduction of cardiorespiratory coupling with high sympathetic tone, the nRSA during elevated anxiety observed in this study might be the reflection of control mechanisms different from cardiorespiratory coupling. It is unclear to me whether the findings of this study agree with this consideration. Discussion should be enlarged to account for this issue.

2)      It is well-known that the latency between respiration (R) and heart period (HP) depends on the respiratory rate (see DL Eckberg, J Appl Physiol, 54, 961-966, 1983). Please report the respiratory rate and prove that the observed phenomenon (i.e., the nRSA) is not the trivial consequence of the faster respiratory rate with elevated anxiety.

3)      Usually, the direction from R to HP was considered because respiratory centers gate autonomic responsiveness and, in turn, affects the sinus node (see A Porta et al, Entropy, 20, 949, 2018). Here the reverse direction of interactions is explored as well. The underlying physiological hypothesis of this analysis should be better explicated.

4)      When analysis the cardiorespiratory coupling from HP to R was carried out, cardiorespiratory coupling from HP to R was found to be significant but autonomic pharmacological challenges did not induce any modification of the strength of this pathway (see A Porta et al, Phil. Trans. R. Soc. A, 371, 20120161, 2013). The authors should discuss the findings of their study in relation to those present in literature.

5)      It remains unclear whether the observed levels of cardiorespiratory coupling strength are significant (i.e., above the levels observed in the case of uncoupled signals). Here the use of a bootstrap approach is reported but no details were given. Given the importance of this part of the analysis for the robustness of final findings I would recommend to the authors to provide more details and describe much better the results of this part of the analysis.

6)      It is unclear whether confounding influences can play a role in the conclusions. Please discuss whether the exploited technique accounted for measured factors that can affect the considered pair of signals and alter the estimation of the relationship between them. In addition, please discuss the possible impact of confounding variables that were not measured in the present experimental setup.

7)      The phrase “In contrast, the classical neurovascular coupling (NVC) corresponds to the hemodynamic BOLD response with dominant CBF and CBV fluctuations and origin in the baroreflex loop (vascular BOLD) [5, 6].” is imprecise given the important contribution of the pressure-to-flow relationship, cerebral autoregulation and Cushing reflex (see V. Bari et al, Physiol Meas, 38, 976-991, 2017).

8)      The authors should better account for studies on directionality of cardiorespiratory coupling present in literature. Section “References” must be improved.

Author Response

Responses to reviewer #2:

We thank this reviewer for his/her valuable and constructive comments.

  • Cardiorespiratory coupling depends on the autonomic nervous system state. Sympathetic activation induced by graded postural challenge reduces cardiorespiratory coupling (see A Porta et al, Comput Biol Med, 42, 298-305, 2012). As a consequence of the reduction of cardiorespiratory coupling with high sympathetic tone, the nRSA during elevated anxiety observed in this study might be the reflection of control mechanisms different from cardiorespiratory coupling. It is unclear to me whether the findings of this study agree with this consideration. Discussion should be enlarged to account for this issue.

Response: We thank this reviewer for his/her valuable comment. We have inserted some sentences on the relation between cardio-respiratory coupling and sympathetic activity, also referring to the recommended literature into the Introduction section.

We have also addressed this issue in the Discussion section. In contrast to various disorders or sympathetic activation, all patterns of respiration-related RRI oscillations observed in the present and in previous studies showed no indication of reduced cardiorespiratory coupling (please see references #29, #30 and #69 in the reference list). One of our methods termed wave-by-wave analysis is a good tool to analyze phase-relations, which are an important feature of coupling. From the results of this wave-by-wave analysis and of the DTF analysis, we would assume that additional rhythmic processes (“pacemaker-like neural activity”) are activated in situations of high anxiety and directly interacts with the coupled rhythms. We have discussed this point in more detail.

  • It is well-known that the latency between respiration (R) and heart period (HP) depends on the respiratory rate (see DL Eckberg, J Appl Physiol, 54, 961-966, 1983). Please report the respiratory rate and prove that the observed phenomenon (i.e., the nRSA) is not the trivial consequence of the faster respiratory rate with elevated anxiety.

Response: A change in the respiratory rate may induce a change in the rate ratio between RRI and respiratory oscillations. However, the wave-by-wave analysis of phase-relations allows to differentiate those changes from true modulations of coupling, which are usually associated with modulations of the phase-relation. We have included this into the Discussion section. At higher respiratory rates, the percentage of RSA or nRSA assessed by wave-by-wave analysis becomes less significant as is discussed in the Limitations section. However, even at higher respiratory rates, the wave-by-wave analysis reveals reliable information on the exact phase-relation between RRI and respiratory waves. We have inserted average respiratory and RRI rates into Table 1.

  • Usually, the direction from R to HP was considered because respiratory centers gate autonomic responsiveness and, in turn, affects the sinus node (see A Porta et al, Entropy, 20, 949, 2018). Here the reverse direction of interactions is explored as well. The underlying physiological hypothesis of this analysis should be better explicated.

Response: We assume that the reversal in the direction of interaction is due to activation of one (or more) additional pacemaker-like activities. Many phenomena observed in the wave-by-wave analysis and the results of the DTF analysis support this hypothesis. We have inserted a more detailed discussion of this issue into the Discussion Section.

  • When analysis the cardiorespiratory coupling from HP to R was carried out, cardiorespiratory coupling from HP to R was found to be significant but autonomic pharmacological challenges did not induce any modification of the strength of this pathway (see A Porta et al, Phil. Trans. R. Soc. A, 371, 20120161, 2013). The authors should discuss the findings of their study in relation to those present in literature.

Response:As we have discussed above, sympathetic activation reduces the HRV magnitude. However, more interactions of autonomic activity and cardiorespiratory coupling can exist. They can, e.g., be part of the hypothesized pacemaker. With respect to the role of autonomic regulation it is interesting that parasympathetic or sympathetic blockade did not modify bidirectional interactions between cardiac and respiratory rhythms. We have inserted this into the Discussion section.

  • It remains unclear whether the observed levels of cardiorespiratory coupling strength are significant (i.e., above the levels observed in the case of uncoupled signals). Here the use of a bootstrap approach is reported but no details were given. Given the importance of this part of the analysis for the robustness of final findings I would recommend to the authors to provide more details and describe much better the results of this part of the analysis.

Response: We have included more details on the bootstrap approach into the Methods section. For a very detailed description of this method we have referred to previous papers where this method has been applied for the first time to our sets of data (references #46 and #58 in the reference list). Indices of significance are presented in the related figure (Fig. 7) and in the text.

  • It is unclear whether confounding influences can play a role in the conclusions. Please discuss whether the exploited technique accounted for measured factors that can affect the considered pair of signals and alter the estimation of the relationship between them. In addition, please discuss the possible impact of confounding variables that were not measured in the present experimental setup.

Response: The wave-by-wave analysis allows exact detection of the phase-relation between the coupled rhythms period by period, which is the advantage of this method. Frequency synchronizations occurring by chance as a consequence of unspecific influences can be detected and excluded. We have previously applied this method in many studies on motor-respiratory control and also in previous studies on RSA. We apologize that a description of this method was lacking in the original manuscript; we have added it in the revised version. The DTF analysis provides a clear information on the significance of outflows and inflows from/to a channel of interest, thus corroborating information about the dominance of a rhythm and the direction of entrainment. Many more factors affecting HRV induce a reduction of the HRV magnitude. This has been discussed in the Discussion section. Unfortunately, we did not analyze the magnitude of HRV in this study, this has been stated in a Limitations section added to the revised version of the manuscript. However, we did not find any hint on a reduced HRV magnitude indicating that those factors may not account for the observed phase-shifts in the cardiorespiratory coupling pattern.

  • The phrase “In contrast, the classical neurovascular coupling (NVC) corresponds to the hemodynamic BOLD response with dominant CBF and CBV fluctuations and origin in the baroreflex loop (vascular BOLD) [5, 6].” is imprecise given the important contribution of the pressure-to-flow relationship, cerebral autoregulation and Cushing reflex (see V. Bari et al, Physiol Meas, 38, 976-991, 2017).

Response: We have changed this sentence as follows: “In contrast, the classical neurovascular coupling (NVC) reflects the close temporal and regional linkage between neural activity and cerebral blood flow (CBF) (Murphy, et al 2013, Huneau et al 2015), whereby cerebral autoregulation, the intrinsic dynamic ability of cerebral vessels to maintain CBF despite fluctuations in arterial blood pressure, plays a dominant role. The evidence of such blood pressure oscillations on BOLD fluctuations is not completely clear and needs further research. In this respect, studies of interaction of cerebrovascular and cardiovascular variability are important (Bari et al. 2017).”

8)      The authors should better account for studies on directionality of cardiorespiratory coupling present in literature. Section “References” must be improved.

Response: We have referred in the Introduction to the literature concerning directionality of cardiorespiratory interaction with proper citations. We have also underlined the difference between directionality and causality.

Round 2

Reviewer 2 Report

The manuscript has been substantially improved. The authors replied satisfactorily to all my issues and took into account the suggestions given. 

Author Response

We thank this reviewer for his/her positive evaluation of our revised manuscript.